# A New Neutralization Epitope in the Spike Protein of Porcine Epidemic Diarrhea Virus

**DOI:** 10.3390/ijms23179674

**Published:** 2022-08-26

**Authors:** Jianbo Liu, Hongyan Shi, Jianfei Chen, Xin Zhang, Da Shi, Zhaoyang Ji, Zhaoyang Jing, Li Feng

**Affiliations:** State Key Laboratory of Veterinary Biotechnology, Harbin Veterinary Research Institute of the Chinese Academy of Agricultural Sciences, 678 Haping Road, Xiangfang District, Harbin 150069, China

**Keywords:** PEDV, neutralizing monoclonal antibody, spike protein

## Abstract

Porcine epidemic diarrhea virus (PEDV) infects piglets and causes serious diarrhea as well as vomiting, dehydration, and death. The trimeric S protein plays a crucial role in the induction of neutralizing antibodies, and many neutralizing monoclonal antibodies (mAbs) against PEDV S protein have been developed. However, these mAbs exclusively target the S1 protein. In this study, we obtained a neutralizing mAb, 5F7, against the S2 protein of PEDV, and this mAb could neutralize new variant genotype 2 PEDV strains (LNCT2), as well as a genotype 1 PEDV strain (CV777), in vitro. The core sequence of the epitope was found in amino acid sequence 1261 aa~1337 aa. These findings confirm that the S2 protein possessed neutralizing epitopes and provided knowledge to aid further research on this virus.

## 1. Introduction

Porcine epidemic diarrhea virus (PEDV) is an enteropathogenic alpha coronavirus belonging to the genus Coronavirus, the family Coronaviridae, and the order Nidovirales. PEDV causes severe enteric illness in swine, infecting newborn and suckling pigs [1], and results in watery diarrhea, vomiting, and death in up to 100% of neonatal piglets [2,3]. PEDV has been prevalent for many years worldwide and has caused enormous economic losses to the swine industry. 

PEDV is an enveloped, nonsegmented, positive-strand RNA virus, the genome of which is 28 kb and contains at least seven open reading frames encoding replicase proteins and four structural proteins: spike (S), membrane, envelope, and nucleocapsid proteins. PEDV strains were divided into two groups based on their genomes: the G1 genotype (including strain CV777) and G2 genotype (including strain LNCT2) [4,5,6]. Subsequently, the G1 genotype was further divided into G1a and G1b, while the G2 genotype was divided into G2a, G2b, and G2c based on the complete genomes of 409 strains of PEDV [7]. The differences between the G1 and G2 strains are mainly found in the S protein, where the S gene of most G2 strains is nine nucleotides longer than that of the prototype PEDV strain CV777 and has many insertions and deletions, as well as other mutations, when compared to the S gene of G1 strains [5]. The S gene has been divided into two segments: S1 and S2, artificially defined based on homology to other coronavirus S proteins [8]. 

The S protein covers the surfaces of the virus in the form of trimers and plays a crucial role in viral entry into the cell and the induction of neutralizing antibodies [9]. Furthermore, the S protein of coronaviruses has been demonstrated to elicit both humoral and cellular immune responses [10,11,12]. Several hybridoma cell strains secreting monoclonal antibodies (mAb) targeting the PEDV S protein have been demonstrated to have neutralizing activity [6,13,14,15]. Some of these mAbs can only neutralize G1 strains, while others neutralize only G2 strains, and some can neutralize both strains. However, most neutralizing mAbs bind to S1, whereas only a few combine with S2. In this study, we attained a neutralizing mAb 5F7 strain that secreted with the ability to bind to the S2 part of the spike protein.

## 2. Results

### 2.1. Positive mAb Clone against PEDV

To develop neutralizing antibodies against PEDV, 6-week-old female BALB/c mice were immunized subcutaneously with purified PEDV to generate mAbs with the methods described by Liu et al. (2017) [6]. After the mice’s spleen cells were fused with SP2/0 myeloma cells and screened, a positive hybridoma cell clone (5F7) that secreted mAbs that reacted strongly with the PEDV strains LNCT2 and CV777 was selected (Figure 1). The heavy chain was IgG2a, and the light chain was kappa, as determined by the SBA Clonotyping System-HRP (Southern Biotech, Birmingham, AL, USA). 

### 2.2. Neutralization Test

Two-fold dilutions of purified mAb 5F7 (10 mg/mL) were used in the neutralization tests. As shown in Figure 2, the half maximal inhibitory concentrations (IC_50_) of mAb 5F7 in neutralizing strains LNCT2 and CV777 were 54.44 μg/mL and 66.17 μg/mL, respectively. These results suggest that mAb 5F7 can neutralize PEDV strains CV777 and LNCT2. 

### 2.3. Identification of the Protein Recognized by mAb 5F7

As shown in Figure 3, the Vero E6 cells in lane 1 were transfected with pcDNA3.1(−)–LNCT2–S, expressing the PEDV S protein, whereas the Vero E6 cells in lanes 2 and 3 were infected with PEDV CV777 and LNCT2, respectively. Vere E6 cells were used as the negative controls. MAb 5F7 reacted with the samples in which the cells were infected with PEDV strain LNCT2 or CV777 and with the cells transfected with pcDNA3.1(−)–LNCT2–S. These results suggest that mAb 5F7 recognized the S protein of the CV777 and LNCT2 strains. 

### 2.4. Identification of the Epitope of mAb 5F7

To identify the epitope of mAb 5F7, different plasmids expressing diverse parts of the S protein were constructed (Figure 4) and used to transfect Vero E6 cells. As shown in Figure 5, the mAb reacted with S2, S2-b, -d, and f but did not react with S1, S2-a, -c, -e, -g, -h, -i, or -j. These results show that the epitope of mAb 5F7 was located in S2-f.

### 2.5. Analysis of the mAb 5F7 Epitope in Other PEDV Strains 

The amino acids in the epitope of mAb 5F7 were highly conserved with only a few amino acid mutations (Figure 6). These results suggest that the epitope is relatively conserved in PEDV.

## 3. Discussion

PEDV is an enteropathogenic swine virus which causes severe enteric disease in newborn and suckling pigs [1]. PEDV is a perennial problem on many pig farms, and the virus has caused substantial economic losses to the swine industry [16]. To control the virus, inactivated and live attenuated vaccines and neutralizing antibodies have been developed [6,14,17], and almost all neutralizing antibodies bind to S1 of the spike protein. Few neutralizing antibodies bind to the S2 protein, except mAb 2C10, the epitope of which is “1368GPRLQPY1374” [18,19]. 

In this study, purified PEDV particles were used to immunize Balb/C and an mAb 5F7 was obtained. The mAb could neutralize not only G1 strains (CV777) but also G2 strains (LNCT2), as shown in Figure 3, and the target protein of the mAb 5F7 was the S protein. To identify the epitope of mAb 5F7, several plasmids were constructed, and finally, the epitope of mAb 5F7 was identified and located in the S2-f (1261 aa~1337 aa) region, and the S2-f peptide was further divided to the S2-h, -I, and -j peptides. Moreover, the S2-h, -I, and -j peptides result in negative results with mAb 5F7. Furthermore, the epitope of 5F7 was neighbored with another epitope, “1368GPRLQPY1374” [18,19]. The epitope of 5F7 occurs at the bottom of the spike trimers and is linked to the carboxyterminal domain, which is involved in the assembly of the spike protein and progeny virions by interacting with the matrix protein [20,21]. The epitope of mAb 5F7 is also similar to the neutralized epitope (the 746ERDRD750 motif) on HIV gp41, which is linked to the end domain [22]. The IC_50_ of mAb 5F7 in neutralizing PEDV was at least 54.44 μg/mL, while the amount of mAb 1B9 required to complete the neutralization of PEDV was 7.813 μg/mL [23]. These results suggest that the neutralization ability of mAb 5F7 is weaker than that of mAb 1B9, and the manner of neutralization of mAb 5F7 may be via the agglutination PEDV or via limiting the rotational freedom of the spikes, which may facilitate the exploring ability of the virus and improve its ability to engage with cellular receptors [24]. However, it is still unclear why antibodies binding to the 1261 aa~1337 aa region reduced virus infectivity in Vero cells and what role this domain had with viral entry. Nevertheless, antibodies that bound the 1261 aa~1337 aa region had neutralizing activities against the virus.

The mAb 5F7 could neutralize strains CV777 and LNCT2, and strain CV777 belonged to the G1 genotype, while LNCT2 belonged to the G2 genotype, and the alignment results suggest that this represented a conserved epitope, except for the two amino acids, T1263I and Q1301R. Because mAb 5F7 could neutralize strains CV777 and LNCT2, and amino acids T1263I and Q1301R occur in strain CV777, these results demonstrate that the two amino acid mutations do not affect the binding of mAb to the epitope. This suggests that these two amino acids are not involved in the conformational epitope. These results therefore demonstrate that the mAb 5F7 is a broad-spectrum neutralizing antibody, and the S2 protein can induce neutralization antibodies. These results provide us with important information to help design subunit vaccines, mRNA vaccines or other gene-engineered subunit vaccines, do not just have to be developments against the S1 or RBD but also against other regions including NTD and S2, which could contain epitopes of neutralizing antibodies [25]. Treatment with more neutralizing antibodies reduces the possibility that a mutant virus will emerge, as demonstrated by Liu et al. (2020) [23]. However, the molecular mechanisms by which mAb 5F7 neutralizes the PEDV strains (such as the exact amino acids involved and the manner of neutralization) and whether the antibodies can be used alone or in combination with other antibodies to cure piglets require further study.

## 4. Materials and Methods

### 4.1. Cells, Virus Propagation, and Purification

Vero E6 cells (CRL-1586; American Type Culture Collection, Manassas, VA, USA) were cultured in our laboratory and used for the PEDV culture. The Vero E6 cell line was maintained in Dulbecco’s Modified Eagle’s medium (DMEM; Gibco BRL Life Technologies, Grand Island, NY, USA) supplemented with 5% fetal bovine serum without antibiotics. The LNCT2 strain and CV777 vaccine strain (GenBank accession nos. KT323980 and KT323979) were cultured in our laboratory. Vero E6 cells were used to propagate PEDV. Virus propagation and purification was as described previously by Liu et al. (2017) [6].

### 4.2. Preparation of mAbs against PEDV

The hybridoma cells methodology was based on standard procedures devised by Köhler and Milstein (1975) [26] and modifications by Liu et al. (2017) [6]. Briefly, 6-week-old female BALB/c mice (from Beijing Vital River Laboratory Animal Technology Co., Ltd., Beijing, China) were immunized subcutaneously with the purified PEDV particles in Freund’s complete adjuvant (Sigma-Aldrich, St. Louis, MO, USA). The mice received two booster immunizations with the PEDV particles in Freund’s incomplete adjuvant at 2-week intervals. Two weeks later, one additional intraperitoneal immunization without adjuvant was administered. Three days after the final dose, the spleen cells of mice were harvested and fused with SP2/0 myeloma cells (store in our lab) at a ratio of 5:1 using polyethylene glycol 1450 (Sigma-Aldrich). The culture and selection of hybridoma cells, as well as the culture supernatants from surviving clones, were screened for reactivity and specificity with an immunofluorescence assay (IFA), as described by Liu et al. (2017) [6], with minor modifications. Briefly, monolayers of Vero E6 cells in 96-well plates were grown to 100% confluence. Then, the cells were inoculated with 1 × 10^2^ PFU per well of the PEDV LNCT2 strain suspended in DMEM supplemented with trypsin. After 48 h, the cells were fixed with 4% paraformaldehyde for 30 min at 4 °C and, after washing with PBS, the cells were incubated with the supernatants of the hybridoma cells for 1 h. Then, they were incubated for 45 min with fluorescein isothiocyanate-labeled goat anti-mouse IgG (Sigma-Aldrich, St. Louis, MO, USA), and the cell layer was examined under a fluorescence microscope. Positive signals suggested that the supernatants from the hybridoma cells were PEDV-specific, IgG-positive mAbs. The positive clones were subcloned three times by the limiting dilution method. All the animal experiments in this study were performed with the approval of our institute, in accordance with animal ethics guidelines and approved protocols.

### 4.3. Antibody Purification and Characterization

The positive clone hybridoma cells were cultured with serum-free medium (Sigma-Aldrich) to obtain their purified antibodies using HiTrap™ Protein G HP column chromatography (GE Healthcare, Uppsala, Sweden) according to the manufacturer’s protocol. The antibody subtypes were detected by the SBA Clonotyping System-HRP (Southern-Biotech, Birmingham, AL, USA) according to the manufacturer’s instructions.

### 4.4. Neutralization Test

A neutralization test was used to determine whether the mAb 5F7 had neutralization activity according to a method described previously by Liu et al. (2017) [6], with some modifications. Briefly, after filtering using a 0.22 μm membrane, the purified mAb (10 mg/mL) was exposed to doubling dilutions to obtain different concentrations. Then, the two-fold dilutions of the mAb (200 μL) were mixed with equal volumes of PEDV containing 200 tissue culture infective doses per 100 μL and incubated for 1 h at 37 °C. After incubation, this mixture, along with trypsin (each milliliter of solution contained 10 μg of trypsin), was added to confluent (100%) monolayers of PEDV negative Vero-E6 cells in four wells of a 96-well plate. Vero E6 cells without antibodies or viruses were used as the negative controls, and the neutralization test result was set to 100%. Then, we observed the cytopathogenic effect (CPE) on these cells for 48 h. PEDV strains (LNCT2 and CV777) were used to perform the neutralization test. The neutralization test result was the number of wells that showed a CPE divided by four and then multiplied by 100%.

### 4.5. Identification of the Target Protein of the mAb by Western Blotting

The plasmid pcDNA3.1(−)–LNCT2-S [6] expressing PEDV S protein was transfected into Vero E6 cells using Lipofectamine 3000 Transfection Kit (Invitrogen, Waltham, MA, USA). The cells were lysed 48 h later and the samples were collected. Vero E6 cells infected by PEDV CV777 and the LNCT2 strain were also lysed 48 h later and collected and Vere E6 cells were used as negative controls. All samples were then analyzed by Western blot. The mAb 5F7 was used as the first antibody and IRDye 800CW goat anti-mouse lgG (H+ L) (LiCor BioSciences, Lincoln, NB, USA) was used as the second antibody. 

### 4.6. Construction of Plasmids and Identification of the mAb Epitope

Various parts of the PEDV S gene were amplified from the plasmid pcDNA3.1(−)–LNCT2–S using the primers listed in Table 1 and the amplicons (S2-a, -b, -c, -d, -e, -f, and -g) were cloned into pcDNA3.1(−) vector (Figure 4). The correct plasmids, pcDNA3.1(−)–LNCT2–S1 and pcDNA3.1(−)–LNCT2–S2 [6], as well as pcDNA3.1(−)–LNCT2–S2-a, -b, -c, -d, -e, -f, -g, -h, -i, and -j, were extracted using a Hipure plasmid filter Miniprep Kit (Invitrogen). The plasmids were transfected into HEK293T cells at a final concentration of 1 μg/mL using Lipofectamine 3000 Transfection Kit (Invitrogen) according to the manufacturer’s instruction, and the empty vector pcDNA3.1(−) was used as the negative control and all cells were fixed using 4% paraformaldehyde. Purified mAb 5F7 was used as the first antibody and fluorescein-isothiocyanate-labeled goat anti-mouse IgG antibody (Sigma-Aldrich) was used as the second antibody. The cell layers were examined under a fluorescence microscope. 

### 4.7. Analysis of the Epitope of mAb 5F7 with Other PEDV Strains

To determine whether the epitope of mAb 5F7 was conserved, the counterpart of the epitope in other PEDV strains (GenBank accession nos. JN547228, KC109141, JQ023162, MK841495, JX088695, and KF468752) with CV777 and LNCT2 were analyzed using the Jotun Hein method in the software program MegAlign.

## 5. Conclusions

In this study, we obtained a hybridoma cell line, 5F7, which secreted PEDV-specific neutralizing antibodies. The 5F7 mAb not only can neutralize PEDV G2 strain (LNCT2) but also neutralize PEDV G1 strain (CV777), and the epitope recognized by mAb 5F7 lay within the S2 protein. The core sequence of the epitope was within 1261 aa~1337 aa, and this 5F7 mAb may be used to treat pigs infected with PEDV at early stages of infection.

## Figures and Tables

**Figure 1 ijms-23-09674-f001:**
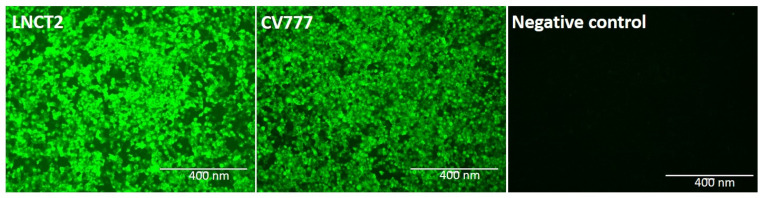
Reaction of mAb 5F7 with PEDV-positive cells, as detected by IFA. The cells infected by PEDV strains LNCT2 and CV777 yielded positive signals, and Vero E6 cells were used as negative controls. The heavy chain of mAb 5F7 was IgG2a, whereas the light chain was kappa, as determined by the SBA Clonotyping System-HRP (Southern Biotech). Bar, 400 μm.

**Figure 2 ijms-23-09674-f002:**
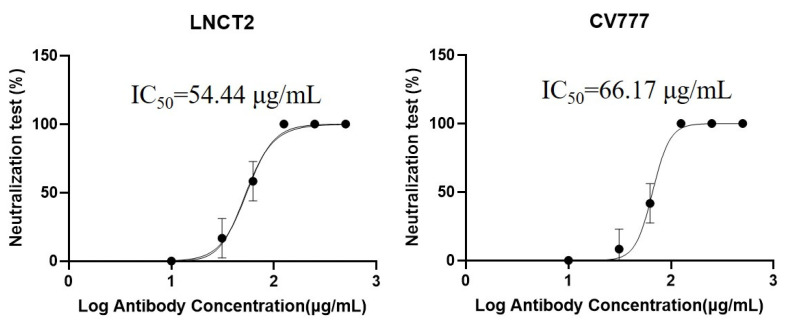
Neutralization of distinct PEDV strains by mAb 5F7. PEDV strains LNCT2 and CV777 were used to test the neutralization features of mAb 5F7. The IC_50_ of the mAb 5F7 in neutralizing strains LNCT2 and CV777 were 54.44 μg/mL and 66.17 μg/mL, respectively. The neutralization tests were repeated three times. The five points of antibody concentration from low to high are 10 μg/mL, 31.25 μg/mL, 62.5 μg/mL 125 μg/mL, 250 μg/mL, 500 μg/mL. The figures were generated with GraphPad Prism version 9.0 (GraphPad Software, La Jolla, CA, USA).

**Figure 3 ijms-23-09674-f003:**
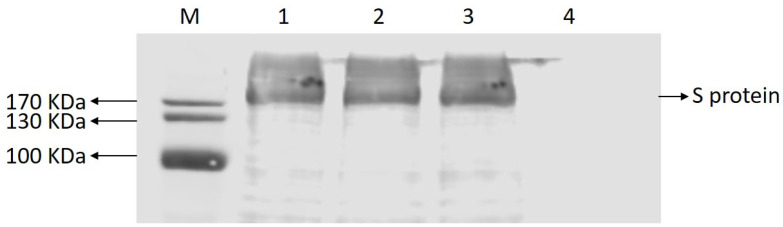
Identification of mAb 5F7 to the PEDV S protein expressed in Vero E6 cells in Western blot analysis. Lane M, protein marker. The Vero E6 cells in lane 1 were transfected with pcDNA3.1(−)–LNCT2–S, expressing the PEDV S protein, whereas the Vero E6 cells in lanes 2 and 3 were infected with PEDV CV777 and LNCT2. Vere E6 cells were used as negative controls (Lane 4). mAb 5F7 reacted well with the PEDV S protein expressed in Vero E6 cells and the S protein of PEDV CV777 and LNCT2. The mAb 5F7 did not react with the Vero E6 cells.

**Figure 4 ijms-23-09674-f004:**
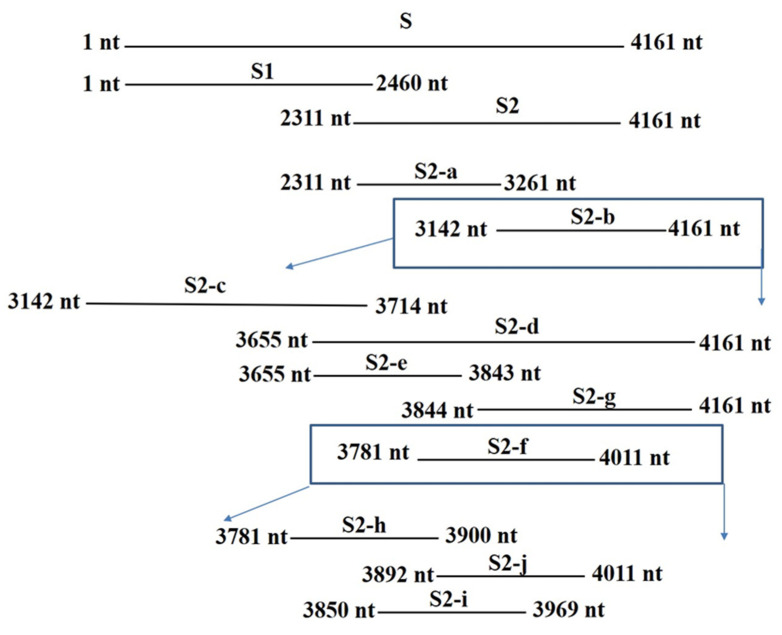
Schematic diagram of the PEDV S fragments used for B-cell epitope mapping. Four rounds of S peptides were conducted to investigate epitopes of the generated mAb.

**Figure 5 ijms-23-09674-f005:**
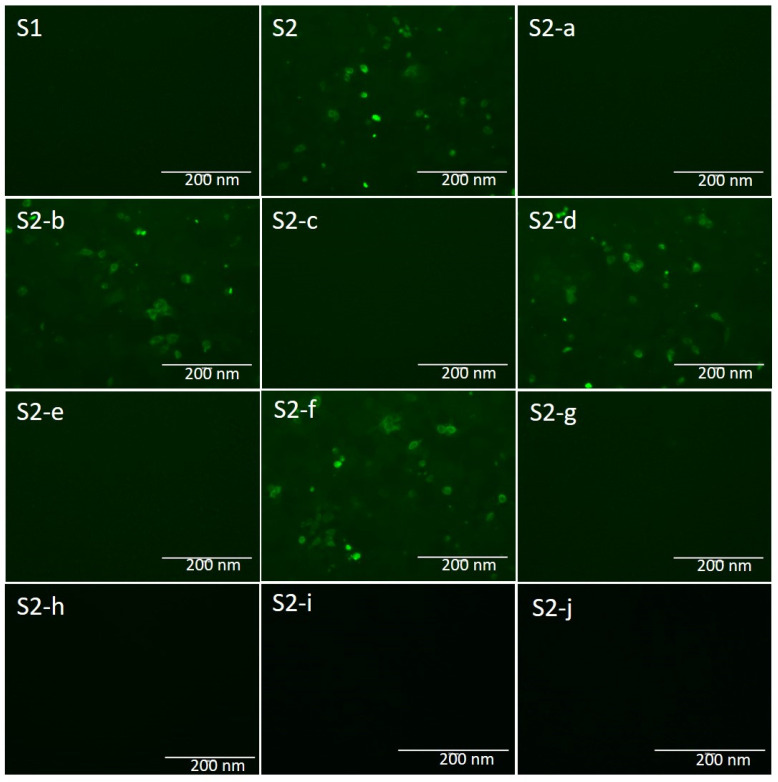
Identification of the epitope recognized by mAb 5F7 using IFA. Recombinant plasmids pcDNA3.1(−)–LNCT2–S1 and –S2, as well as pcDNA3.1(−)–LNCT2–S2-a, -b, -c, -d, -e, -f, and -g, were used to transfect 293T cells. The results show that the cells transfected with pcDNA3.1(−)–LNCT2-S2, S2-b, -d, and f plasmids reacted with mAb 5F7, while the cells transfected with pcDNA3.1(−)–LNCT2-S1, S2-a, -c, -e, -g, -h, -i, and -j plasmids did not react with mAb 5F7. Bar, 200 μm.

**Figure 6 ijms-23-09674-f006:**
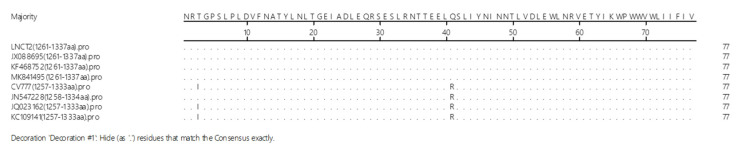
Comparison of the mAb 5F7 epitope amino acid sequence among the different PEDV strains. The mAb 5F7 epitope sequences from different strains of PEDV are shown.

**Table 1 ijms-23-09674-t001:** Oligonucleotide Primers Used for Amplified PEDV S Gene.

Primers	Sequence
F2311	5’-AGACTCGAGATGCAAGTGAAGATCGCCCCTACCGTGACC-3’
F3142	5’-AGACTCGAGATGGCCCTGACACAGCTGACAGTGC -3’
F3655	5’-AGACTCGAGATGGACTTCGTGCAGATTGAGAGCTGCGTC-3’
F3781	5’-AGACTCGAGATGAACAGAACCGGCCCTTCTCTGC-3’
F3844	5’-AGACTCGAGATGGAGATCGCCGACCTGGAACAGAGAAG-3’
F3850	5’-AGACTCGAGATGGCCGACCTGGAACAGAGAAGCGAGAG-3’
F3892	5’-AGACTCGAGATGGAGGAACTGCAGAGCCTGATCTATAA-3’
R3261	5’-GTGGAATTCTTAGATCAGCCGGTCGATCTGCACGT-3’
R3714	5’-GTGGAATTCTTACAGCTGGTCCCTGGTCAGGTT-3’
R3843	5’-GTGGAATTCTTAGCCGGTCAGATTCAGGTATGTGGC-3’
R3900	5’-GTGGAATTCTTACAGTTCCTCGGTGGTGTTCC-3’
R3969	5’-GTGGAATTCTTAATAGGTTTCCACCCGGTTCAGCC-3’
R4011	5’-GTGGAATTCTTACACGATAAAGATGATCAGCCACA-3’
R4161	5’-GTGGAATTCTTATCACTGCACATGCACTTTCTCG-3’

## Data Availability

Source data for figures are provided in the article. All reagents generated in this study are available upon reasonable request. Source data are provided with this paper.

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
