# Peer review of "A New Neutralization Epitope in the Spike Protein of Porcine Epidemic Diarrhea Virus"

_ijms, 2022, doi:10.3390/ijms23179674_

Round 1
Reviewer 1 Report
In this manuscript, Liu et al., reported about new neutralization epitope against PEDV. They showed that novel S2 epitopes of PEDV potently neutralized the virus. The finding is interesting and important for the improvement of novel therapeutic antibody. However, the manuscript lack some
Information including the explanations about their data and hard to understand. The authors need to improve the manuscript. The following is my concerns.
Major
1. Line 52: “Mice were immunized with…” The authors need to add detailed information (at least mouse strain and how to immunized mice) in the Material and Method.
2. Regarding “5F7”, what is the source of this? Did the authors identify it? They need to explain more about this in the Introduction.
3. Figure1 : Please change “E6” to “Uninfected” or “negative control”.
4. Line 55-56: this information must put in the figure legend.
5. Section 2.1: The authors need to explain more about their data. They need to start from their purpose of these experiments and describe more how they conducted the experiments.
6. Figure 1 legend: Please add the information about bar.
7. Section 2.2: The authors need to improve this section. They need to at least explain more about their data. For example, they need to explain how fold the antibody decreased the infectivity and need to add the conclusion.
8. Line 63-64: The authors described that “the LNCT2, and CV777 strains were neutralized by mAb 5F7 at concentrations of 62.5 μg/mL and 125 μg/mL, respectively.” But based on their data, it is not correct. 500-125ug/ml Abs neutralized the virus completely. Please fix the sentence.
9. Figure 2: How the authors calculate (%)? Do the authors set “no Abs” as 100%? This information must add in the Material and Method.
10. Figure 2: the authors might need to use much narrow (125-62.5ug/ml) and lower concentration of Abs and calculate IC50 (ug/ml).
11. Figure 2: The authors should perform the statics analysis. They also add “n” numbers in the figure legend.
12. Figure 2: “Antibody dilution” should be “Antibody Concentration”.
13. line 67-69. This sentence should be add in the main section, not legend.
14. Figure 3: The S protein band looks cut off. Please fix them.
15. Line 79-81: this sentence must add in the main section, not legend.
16. Section 2.4. the authors need to explain more. Do the authors calculate % of green cells? This information must be important to accurately identify the location.
17. Figure 6 must be added before Figure 4. Without the information of Figure 6, it is hard to understand what “S1-S2-j” means.
18. Figure 5: The mutation located in 41aa, do the authors think this mutation does not affect the neutralizing activity? The authors need to describe/discuss in the Result or Discussion.
19. In the Discussion, the authors need to discuss more how the 5F7 Ab will be apply to the therapeutic use.
Author Response
Response: Thank the reviewer very much for this comment. The reviewer’s suggestions are very good for this manuscript. All the changes were listed in the revision as follow. Some sentences that need to be improved have been revised and marked in yellow.
- Line 52: “Mice were immunized with…” The authors need to add detailed information (at least mouse strain and how to immunized mice) in the Material and Method.
Response: The detailed information about mouse strain and how to immunized mice has been added in Line 52.
- Regarding “5F7”, what is the source of this? Did the authors identify it? They need to explain more about this in the Introduction.
Response: The mAb 5F7 was identified in this study, and the information has been added in the Introduction.
- Figure 1: Please change “E6” to “Uninfected” or “negative control”.
Response: “E6” has been changed into “negative control”.
- Line 55-56: this information must put in the figure legend.
Response: The information in Line 55-56 has been put in the figure legend.
- Section 2.1: The authors need to explain more about their data. They need to start from their purpose of these experiments and describe more how they conducted the experiments.
Response: The purpose of the experiments and the details of the experiments has been added in Section 2.1.
- Figure 1 legend: Please add the information about bar.
Response: The information about bar has been added in Figure 1 legend.
- Section 2.2: The authors need to improve this section. They need to at least explain more about their data. For example, they need to explain how fold the antibody decreased the infectivity and need to add the conclusion.
Response: The Section 2.2 has been improved.
- Line 63-64: The authors described that “the LNCT2, and CV777 strains were neutralized by mAb 5F7 at concentrations of 62.5 μg/mL and 125 μg/mL, respectively.” But based on their data, it is not correct. 500-125ug/ml Abs neutralized the virus completely. Please fix the sentence.
Response: The sentence has been fixed.
- Figure 2: How the authors calculate (%)? Do the authors set “no Abs” as 100%? This information must add in the Material and Method.
Response: This information has been added in the Material and Method.
- Figure 2: the authors might need to use much narrow (125-62.5ug/ml) and lower concentration of Abs and calculate IC50 (ug/ml).
Response: The IC50 was calculate by GraphPad Prism 9.0.0.
- Figure 2: The authors should perform the statics analysis. They also add “n” numbers in the figure legend.
Response: The numbers was added in the figure legend.
- Figure 2: “Antibody dilution” should be “Antibody Concentration”.
Response: “Antibody dilution” has been changed into “Antibody Concentration”.
- line 67-69. This sentence should be add in the main section, not legend.
Response: The sentence has been added in the main section
- Figure 3: The S protein band looks cut off. Please fix them.
Response: We have changed the Figure 3.
- Line 79-81: this sentence must add in the main section, not legend.
Response: The sentence has been added into the main section.
- Section 2.4. the authors need to explain more. Do the authors calculate % of green cells? This information must be important to accurately identify the location.
Response: The green cells were used to show that the protein expressed could bind by mAb 5F7. The information of S2- a, b, c, d, e, f, g, h, i, and j was shown in Figure 4 which was the former Figure 6.
- Figure 6 must be added before Figure 4. Without the information of Figure 6, it is hard to understand what “S1-S2-j” means.
Response: Figure 6 has been added before Figure 4.
- Figure 5: The mutation located in 41aa, do the authors think this mutation does not affect the neutralizing activity? The authors need to describe/discuss in the Result or Discussion. Response: Discussion about the mutant amino acid in the epitope of mAb 5F7 has been added in the Discussion.
- In the Discussion, the authors need to discuss more how the 5F7 Ab will be apply to the therapeutic use.
Response: Discussion about the therapeutic use of mAb 5F7 has been added into the Discussion section.
Reviewer 2 Report
The manuscript entitled: “A new neutralization epitope in the spike protein of porcine epidemic diarrhea virus (ID: ijms-1859079) by Jianbo Liu et al. attained a neutralizing 48 mAb strain which had the ability to bind to the S2 part of the spike protein.
Albeit the manuscript is well prepared and a special interest, some minor comment should be addressed.
1. The authors should clarify why SP2/0 myeloma cells were used?
2. Figure 5 should be enlarged.
3. Discussion and conclusion section: The authors should highlight in more detail further potential unmet need in further investigations and clinical implications.
Author Response
Response: Thank the reviewer very much for this comment. The reviewer’s suggestions are very good for this manuscript. All the changes were listed in the revision as follow. Some sentences that need to be improved have been revised and marked in yellow. And the English language has been revised by International Science Editing.
- The authors should clarify why SP2/0 myeloma cells were used?
Response: In this study, the SP2/0 myeloma cells were used to gain cell lines that secreting antibodies because this is the classical and easiest methods to gain monoclonal antibody.
- Figure 5 should be enlarged.
Response: The Figure has been enlarged.
- Discussion and conclusion section: The authors should highlight in more detail further potential unmet need in further investigations and clinical implications.
Response:Potential unmet need in further investigations and clinical implications has been highlighted in the Discussion section.
Round 2
Reviewer 1 Report
In this manuscript, Liu et al., sought to find novel epitopes against porcine epidemic diarrhea virus. They showed that the core sequence located in S2 protein was the one of the main/important epitopes. These findings might be important for the development of new vaccines/drugs against this virus. In this revised manuscript, they added some data and improved it. Now the manuscript became better, but I still have a few comments for the publication.
Minor
1. Line 52-53: “6-week-old BALB/c mice were…” This method and source of mice should be added in the Material and Method.
2. Line 54-55: “SP2/0 myeloma cells” The source of this cell should be added in the Material and Method.
3. Section 2.2: The authors showed that IC50 of mAb 5F7 are 54 and 66 ug/mL against 2-strains. Do the authors think this mAb has strong activity against both strains? They should conclude something in the section 2.2.
4. Figure 3: The information about lane 4 is missing. Please add “lane4” in line 86 (after negative control).
5. Line 173: “1x102” should be “1x102”.
Author Response
Response: Thank the reviewer very much for this comment. The reviewer’s suggestions are very good for this manuscript. All the changes were listed in the revision as follow. Some sentences that need to be improved have been revised and marked in yellow. The language has been revised by International Science Editing.
- Line 52-53: “6-week-old BALB/c mice were…” This method and source of mice should be added in the Material and Method.
Response: The method and source of mice have been added in the Material and Method.
- Line 54-55: “SP2/0 myeloma cells” The source of this cell should be added in the Material and Method.
Response: The source of SP2/0 myeloma cells was added in the Material and Method.
- Section 2.2: The authors showed that IC50 of mAb 5F7 are 54 and 66 ug/mL against 2-strains. Do the authors think this mAb has strong activity against both strains? They should conclude something in the section 2.2.
Response: “These results suggest that mAb 5F7 can neutralize PEDV strains CV777 and LNCT2” has been added in the section 2.2.
- Figure 3: The information about lane 4 is missing. Please add “lane4” in line 86 (after negative control).
Response: “lane 4” in line 86 was added after negative control.
- Line 173: “1x102” should be “1x102”.
Response: 1x102 has been changed into 1x102.